# Hierarchical Shortest-Path Graph Kernel Network

**Jiaxin Wang**[1]    **Wenxuan Tu**[2*]   **Jieren Cheng**[1,2*]
[1]School of Cyberspace Security, Hainan University
[2]School of Computer Science and Technology, Hainan University
{twx,992730}@hainanu.edu.cn

## Abstract

Graph kernels have emerged as a fundamental and widely adopted technique in graph machine learning. However, most existing graph kernel methods rely on fixed graph similarity estimation that cannot be directly optimized for task-specific objectives, leading to sub-optimal performance. To address this limitation, we propose a kernel-based learning framework called Hierarchical Shortest-Path Graph Kernel Network (**HSP-GKN**), which seamlessly integrates graph similarity estimation with downstream tasks within a unified optimization framework. Specifically, we design a hierarchical shortest-path graph kernel that efficiently preserves both the semantic and structural information of a given graph by transforming it into hierarchical features used for subsequent neural network learning. Building upon this kernel, we develop a novel end-to-end learning framework that matches hierarchical graph features with learnable *hidden* graph features to produce a similarity vector. This similarity vector subsequently serves as the graph embedding for end-to-end training, enabling the neural network to learn task-specific representations. Extensive experimental results demonstrate the effectiveness and superiority of the designed kernel and its corresponding learning framework compared to current competitors. Code is available at `https://github.com/JXWANG-GRAPH/HSP-GKN`.

## 1   Introduction

Graph neural networks (GNNs) have become a prominent approach in graph machine learning [1, 2, 3, 4, 5, 6, 7, 8, 9]. Most GNN models belong to the family of message-passing neural networks, where node representations are updated by passing messages and aggregating information from neighboring nodes [10, 11, 12, 13, 14, 15]. However, existing GNNs typically use permutation-invariant readout functions to aggregate the node representations into a graph embedding. Recent studies have shown that their expressive power is constrained by the 1-Weisfeiler-Lehman (1-WL) test [16, 17], which limits their ability to capture complex graph structures and affects their performance on graph-level tasks [18, 19, 20].

Thanks to the ability to model complex relationships, graph kernel methods have been powerful tools for analyzing and learning from graph-structured data [21]. By employing mechanisms such as substructure matching and topological feature extraction, these graph kernels have been able to effectively capture the structural information of graphs [22]. Despite their great successes, graph kernels rely on handcrafted kernel functions, which pose challenges in adapting flexibly to diverse task requirements. On the one hand, most graph kernel methods measure graph similarity in an implicit manner, thereby limiting their ability to incorporate with neural networks for learning task-relevant representations. On the other hand, certain kernel methods, such as the Neighborhood Hash Kernel [23] and Graphlet [24], employ the approach of explicit feature mappings. However, these methods typically represent graphs using the frequency of substructure occurrences or symbolic

---

*Corresponding author.

labels, ignoring the semantic information crucial for accurate graph similarity matching [25]. Unlike kernel methods, neural networks are capable of extracting task-relevant features from the data driven by the objective function. Moreover, the high computational complexity of kernel methods renders them impractical for large-scale datasets. Consequently, it is critically important to develop a novel framework that combines the structural feature-capturing advantages of graph kernels with the representation learning capabilities of neural networks.

An intuitive solution is to design a framework that facilitates the negotiation between the graph kernel learning and neural network optimization processes. Within this framework, a kernel serves as a measurement of graph similarity, while the neural network is incorporated to integrate this process with the task objective for task-specific representation learning. To fulfill this, there are two key challenges to be addressed, i.e., 1) the selected kernel should capture both the semantic and structural information of the graph; and 2) the similarity measurement process driven by the kernel is supposed to be differentiable, enabling the network to be end-to-end trainable. For the first challenge, inspired by the findings from previous work [26, 27] that distance encoding is more expressive than the 1-WL algorithm, we attempt to

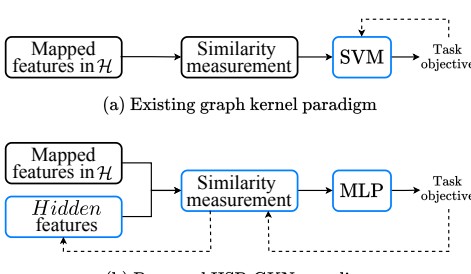

(a) Existing graph kernel paradigm

(b) Proposed HSP-GKN paradigm

Figure 1: Optimization pipeline comparison of existing graph kernel methods and proposed HSP-GKN framework.

design the kernel based on the shortest paths. Since the nodes on the shortest paths provide rich semantic information, the positions of different nodes on the shortest path reflect the topological information of the path. For the second challenge, we explicitly map the graph into a vector space where similarity can be measured using the vector inner product. Since the vector inner product is differentiable, it facilitates the joint learning between kernel-induced features and the neural network to obtain more useful representations for specific tasks [28].

In this paper, we propose a simple yet effective kernel-based graph learning framework, termed Hierarchical Shortest-Path Graph Kernel Network (HSP-GKN). Specifically, we first design a computationally efficient graph kernel based on the shortest paths in the graph, which constructs an explicit feature mapping that transforms the graph into hierarchical features to preserve both the semantic and structural information of the given graph. Next, we integrate task objective and similarity measurement into a unified optimization framework. In this framework, we first define a set of trainable *hidden* graph features and then use the designed kernel to measure their similarity with the hierarchical features. The resultant similarity vector serves as the graph embedding that is fed into a multilayer perceptron (MLP) for training. Unlike previous graph kernel methods that isolate the learning processes of graph representations and objective optimization [29], the proposed HSP-GKN integrates them into a unified framework to learn representations that are more relevant to specific tasks, as illustrated in Figure 1. The dashed lines represent the optimization process of the task objective, with the blue borders indicating trainable components and $\mathcal{H}$ denoting a Hilbert space.

Our contributions include: (1) We design a computationally efficient kernel based on the shortest paths. It can be flexibly integrated with neural networks while preserving both the semantic and structural information of a given graph. (2) We develop a novel framework termed HSP-GKN which integrates the kernel we proposed with neural networks, enabling objective-driven learning of representations beneficial for specific tasks. (3) Experimental results on graph classification datasets across various domains demonstrate the proposed kernel and framework outperform traditional graph kernel methods and state-of-the-art models in terms of both performance and computational efficiency.

## 2   Related Work

**Graph kernels.** Most graph kernels follow the R-convolution framework [30], which decomposes graphs into substructures and computes similarities via kernel functions, aggregating them to yield the overall graph similarity. Different graph kernels are computed in distinct ways, we categorize them into explicitly and implicitly computed graph kernels in this work. The idea of explicitly computing graph kernels involves mapping a graph into a feature vector and measuring the similarity

between the feature vectors of two graphs [23, 24, 31, 32]. These methods typically map graphs into feature vectors composed of the frequency of substructure occurrences, focusing mainly on graph topology while neglecting node semantic information. Unlike these methods, implicit computed kernels measure similarity between substructures that are hard to represent explicitly. The RW kernel [33] is one of the first methods that implicitly measures graph similarity based on the number of common walks between two graphs. Unlike random walks, shortest paths capture both local and global structures, while node attributes along paths provide rich semantics, motivating various shortest-path-based kernels [31, 34, 35, 36]. Despite their success, the implicit similarity computation limits the ability to integrate neural networks for learning task-specific representations. Moreover, the high computational complexity (e.g., SP kernel [34] has a complexity of $\mathcal{O}(n^4)$ with respect to the sample size $n$) restricts their applicability to large-scale datasets. In this paper, we design a graph kernel with a computational complexity of $\mathcal{O}(n^2 \log n)$. It employs explicit computation, allowing the similarity measurement process to be optimized end-to-end, facilitating both model efficiency and task-specific learning.

**Graph kernel neural networks (GKNNs).** Recent research has explored the integration of graph kernels and neural networks, leading to notable performance improvements [37, 38, 39, 40]. For example, RWNN [37] introduces random walk-based mechanisms to capture higher-order dependencies and improve node representation by incorporating both local and global structural information. GNN-lofi [39] leverages localized feature-based histogram intersection to enhance the representation learning of graph by focusing on local feature distribution matching. The success of GKNNs lies in their ability to combine the complementary strengths of graph kernels and neural networks [38, 41]. Kernel functions allow GKNNs to effectively capture structural similarities between graphs, while neural networks provide the capacity to learn complex and abstract graph representations. Despite their success, GKNNs still face several challenges. Scalability remains a major concern, particularly when applied to large-scale graph datasets. Moreover, existing methods either rely on implicitly computed kernels, which prevent the model from learning task-relevant representations, or focus solely on substructures while overlooking rich semantic information in graphs. In this paper, we construct an computationally efficient explicit feature mapping that incorporates both structural and semantic information, mapping graphs into vector space amenable to neural network processing.

## 3 Methodology

In this section, we first introduce key notations employed throughout the paper. Next, we describe the proposed Hierarchical Shortest-Path (HSP) graph kernel, which serves as the theoretical foundation of our framework. Finally, we present the end-to-end learning framework, i.e., Hierarchical Shortest-Path Graph Kernel Networks (HSP-GKN).

### 3.1 Notations

Let $G = (V, E, a)$ be an undirected graph, where $V$ is the vertex set and $E$ is the edge set, and $a\colon V \to \Sigma$ is a function that assigns attributes, either discrete or continuous, from a set $\Sigma$ to nodes in the graph. Given two nodes $v_a, v_b \in V$, a path $\pi$ of length $n-1$ from $v_a$ to $v_b$ in $G$ is defined as a sequence of nodes $\pi = [v_1, v_2, \ldots, v_n]$, where $v_1 = v_a$, $v_n = v_b$, and $[v_i, v_{i+1}] \in E$ for all $i = 1, \ldots, n-1$. Let $\pi(i) = v_i$ denote the $i$-th node encountered while traversing along the path. In particular, we denote by $a(\pi)$ the concatenation of node attributes along this path. Let $|\pi|$ be the discrete length of $\pi$, and the diameter $\delta(G)$ of $G$ is the maximum length of the shortest path between any two nodes in the graph. We use $\mathbf{X} \in \mathbb{R}^{n \times d}$ to denote the node attribute matrix of a graph, where $n$ is the number of nodes and $d$ is the dimension of node attribute. The attribute of a given node $v_i$ corresponds to the $i$-th row of $\mathbf{X}$.

### 3.2 Hierarchical Shortest-Path Graph Kernel

In this subsection, we introduce the Hierarchical Shortest Path-Graph Kernel, which encodes both the semantic and topological information of graphs into explicit features and describes an efficient method for its computation.

**Definition 3.1. GraphHopper (GH) kernel** [36]. Given two graphs, $G$ and $G'$, let $\mathcal{SP}_l(G)$ and $\mathcal{SP}_l(G')$ denote the set of all shortest paths of length $l$ in graph $G$ and $G'$. We can then define the

GH kernel as:

$$k\left(G, G'\right) = \sum_{l=1}^{L} \left[ \sum_{\pi \in \mathcal{SP}_l(G)} \sum_{\pi' \in \mathcal{SP}_l(G')} k_{path}\left(\pi, \pi'\right) \right], \tag{1}$$

where $L$ denotes the results of $min\left(\delta\left(G\right), \delta\left(G'\right)\right)$, and $k_{path}$ the sum of the kernel over all nodes along the shortest paths.

However, all shortest paths need to be compared in GH kernel, and a graph can have at most $|V|^2$ shortest paths, the computational complexity of $k(G, G')$ results in $\mathcal{O}(|V|^2|V'|^2)$. This makes computation infeasible for large graphs. Moreover, like other kernels based on shortest paths, the GH kernel performs implicit computation, resulting in a computational complexity that is quadratic in relation to the number of graphs. Consequently, we design a computationally efficient instance of the GH kernel, referred to as the **Hierarchical Shortest-Path (HSP) Graph Kernel**.

In the HSP kernel, we restrict the node-level kernel $k_{node}$ to a linear kernel, that is:

$$k_{path}(\pi, \pi') = \sum_{i=1}^{|\pi|+1} k_{node}\left(\pi\left(i\right), \pi'\left(i\right)\right), \text{ with } k_{node}(v, v') = \langle a\left(v\right), a\left(v'\right)\rangle. \tag{2}$$

Since the linear kernel is positive definite and the sum of linear kernels is still a linear kernel, $k_{path}$ remains a linear kernel and is also positive definite. By exploiting the linear additivity property of the linear kernel, we construct a hierarchical feature mapping that aggregates the set of shortest paths of the same length into a *cohesive shortest-path*.

### 3.3 Hierarchical Graph Feature Mapping

Differing from previous graph feature mapping, which maps the graph to a fixed-dimensional feature, we transform the graph into a feature set organized by the shortest-path length. The hierarchical graph feature mapping steps are grouping all shortest paths of the same length and aggregating the node attributes at the corresponding indices along the shortest paths, therefore preserving the semantic information of nodes and their relative positions along the shortest-path.

**Definition 3.2.** Given a graph $G$, let $\mathcal{SP}_l(G)$ denote the set of shortest paths of length $l$ in $G$. We define the hierarchical graph feature mapping as:

$$\Phi(G) = \{\phi_l(G) \mid l = 1, 2, \ldots, \delta(G)\}, \text{ with } \phi_l(G) = \sum_{\pi \in \mathcal{SP}_l(G)} a(\pi). \tag{3}$$

This formula aggregates the shortest paths of the same length into a single *cohesive shortest-path*, and we can view the graph $G$ as one that contains only $\delta(G)$ shortest paths. This approach eliminates the need to compare each pair of shortest paths of the same length between the two graphs, thereby significantly reducing computational cost. We can now simplify Eq. (1) as follows:

$$k\left(G, G'\right) = \sum_{l=1}^{L} \langle \phi_l\left(G\right), \phi_l\left(G'\right)\rangle, \tag{4}$$

where $L$ denotes the results of $min\left(\delta\left(G\right), \delta\left(G'\right)\right)$. The derivation of Eq. (4) see Appendix A.1. In this case, the computational complexity of $k\left(G, G'\right)$ is $\mathcal{O}\left(\delta(G)^2\delta(G')^2\right)$, where $\delta(G)$ is the diameter of graph $G$, in real-world graphs $\delta(G)$ is typically much smaller than $|V|$. However, computing $\Phi(G)$ requires at most $|V|^2$ addition operations on $a(\pi)$, which remains a high-complexity operation. In the following, we further discuss how to compute $\Phi(G)$ efficiently.

Given the node attribute matrix $\mathbf{X}$ and the set of all shortest paths of length $l$ in the graph $G$, denoted as $\mathcal{SP}_l(G)$. $\phi_l(G)$ essentially represents the weighted sum of all nodes on the shortest paths of length $l$. Therefore, when searching for the shortest paths in the graph, we simultaneously maintain a matrix $\mathbf{M}_l \in \mathbb{R}^{n \times (l+1)}$, where the entry $[\mathbf{M}_l]_{ij}$ counts how many times node $i$ appears at the $j$-th coordinate of the shortest paths in $\mathcal{SP}_l(G)$. To avoid confusion, it is worth noting that $M_l$ is different from $M(v)$ as defined in the GH kernel, where $[M(v)]_{i,j}$ represents the number of times $v$ appears as the $i^{th}$ node on a shortest path of discrete length $j$. From this, it is straightforward to derive that:

$$\phi_l(G) = \text{vec}(\mathbf{X}^\top \mathbf{M}_l), \tag{5}$$

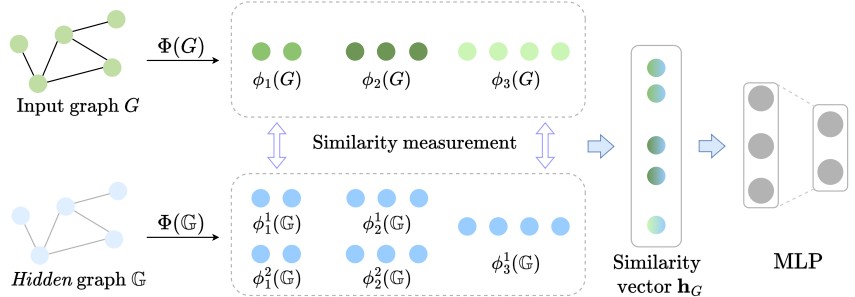

Figure 2: An overview of HSP-GKN. $\Phi(\cdot)$ is the hierarchical graph feature mapping.

where $\text{vec}(\cdot)$ denotes the vectorization operator, which transforms a matrix into a vector by stacking the columns of the matrix one after another. Now, computing $\Phi(G)$ only need to perform $\delta(G) \times |V|$ multiplication and addition operations, which is much smaller than the $|V|^2$ additions required.

**Explicit feature mapping.** It is known that if a positive definite kernel $k(\cdot, \cdot)$ is defined on $\mathcal{X} \times \mathcal{X}$, there exists a mapping $\phi : \mathcal{X} \to \mathcal{H}$ that maps the set $\mathcal{X}$ into a Hilbert space $\mathcal{H}$ with inner product $\langle \cdot, \cdot \rangle$, such that for $\forall x_i, x_j \in \mathcal{X}$, it holds that:

$$k\left(x_i, x_j\right) = \left\langle \phi\left(x_i\right), \phi\left(x_j\right)\right\rangle_{\mathcal{H}}. \tag{6}$$

Therefore, for a positive definite kernel, there exists a feature space where similarity can be computed using an inner product. However, the hierarchical feature mapping that we define represents graphs in the form of sets, which cannot be computed using standard inner products. Therefore, the set-based representation does not constitute the feature space associated with our kernel.

Since our goal is to construct an explicit feature mapping that transforms graphs into vector representations suitable for neural network processing, we next explore how to explicitly map this set into a vector space corresponding to the HSP kernel.

**Lemma 3.1.** *The hierarchical shortest-path graph kernel is positive definite.*

**Lemma 3.2.** *If a graph contains the shortest path of length $l$, then the shortest paths of length $1, 2, \ldots, l-1$ must also exist.*

The proof of Lemma 3.1 and Lemma 3.2 can be found in Appendix A.2. According to Lemma 3.2, we can conclude that although some graphs have different diameters, sorting the shortest paths of two graphs in ascending order of length allows their path lengths to overlap partially within a certain range. Now, we arrange $\phi_l(G)$ into an ordered set by sorting according to the corresponding shortest-path length $l$ in ascending order. Subsequently, by applying a concatenation operation, we obtain the feature vector representation of $G$:

$$\Phi(G) = \text{CONCAT}\left(\phi_1(G), \phi_2(G), \ldots, \phi_{\delta(G)}(G)\right), \tag{7}$$

where $\text{CONCAT}(\cdot)$ is the vertical concatenation of vectors. It follows that $\phi_l(G)$ and $\phi_l(G')$ have corresponding positions in $\Phi(G)$ and $\Phi(G')$. Let $\odot$ denote the aligned inner product of vectors of possibly different lengths, e.g.

$$[a, b, c] \odot [d, e] = [a, b] \cdot [d, e]. \tag{8}$$

Referring to Eq. (4), we obtain:

$$k\left(G, G'\right) = \Phi(G) \odot \Phi(G'). \tag{9}$$

We now further discuss the feature vector representation of $G$. If we zero-extend $\Phi(G)$ to a sufficiently large dimension that exceeds the dimension of any possible graph, Eq. (9) can be extended to the standard inner product computation. Therefore, the extended vector is essentially the mapped feature vector induced by the HSP kernel in the feature space $\mathcal{H}$.

### 3.4 Graph Kernel Network Based on HSP

In this subsection, we present our proposed learning framework, HSP-GKN, and explain how to integrate the similarity measurement of the HSP kernel with objective optimization into a unified framework. An overview of HSP-GKN is illustrated in Figure 2.

As defined previously, hierarchical graph feature mapping transforms a graph into a set of graph features, referred to as hierarchical features. The main steps of HSP-GKN begin with defining a set of learnable *hidden* graph features. Then, a similarity vector is obtained by measuring the similarity between these learnable features and the hierarchical features. Finally, we feed this similarity vector as the graph embedding into MLP and optimize the framework via an objective function.

**Hidden graph features.** From Eq. (9), we know that the similarity measurement process in HSP kernels is non-learnable, as each graph is mapped to a fixed vector representation. Inspired by RWNN [37], we extend the concept of *hidden* graph features, i.e., parameterized graph features into paths. Specifically, the set of *hidden* graph features is defined as:

$$\{\phi_l^{(i)}(\mathbb{G}) \mid i = 1, \ldots, n_l;\ l = 1, \ldots, L\},\ \text{with}\ n_1 + n_2 + \cdots + n_L = N. \tag{10}$$

Here, $L$ and $N$ are hyperparameters to adjust the hierarchy and number of *hidden* graph features, $n_l$ and the number of shortest paths of length $l$ in all the graphs of the training set are proportional. It is worth noting that $\mathbb{G}$ is not an actual graph; instead, it was introduced to facilitate the definition. We can consider this set as consisting of features derived from $N$ learnable graphs, each obtained through the feature mapping.

**End-to-end optimization framework.** After defining the *hidden* graph features, we use the linear kernel used by the HSP kernel to measure the similarity between graph $G$ and these features, resulting in a similarity vector:

$$\mathbf{h}_G = [\left\langle \phi_l(G), \phi_l^{(i)}(\mathbb{G}) \right\rangle \mid i = 1, \ldots, n_l;\ l = 1, \ldots, L]. \tag{11}$$

In what follows, we provide more details about the implementation of the proposed HSP-GKN framework.

Given an input graph $G$, we first obtain the hierarchical feature vector representations set of the graph $\{\phi_l(G) \mid l = 1, 2, \ldots, \delta(G)\}$ through Eq. (5). Then, we define a set of trainable parameters that correspond to the set of *hidden* graph features:

$$\mathcal{W} = \{\mathbf{W}_1, \mathbf{W}_2, \ldots, \mathbf{W}_L\}, \tag{12}$$

where $\mathbf{W}_l \in \mathbb{R}^{[(l+1) \times d] \times n_l}$, and $d$ is the dimension of the node attributes. The $i$-th column of $\mathbf{W}_l$ represents the vector representation of $\phi_l^{(i)}(\mathbb{G})$. Subsequently, we can obtain the similarity vector between the graph $G$ and the feature vectors:

$$\mathbf{h}_G = [\phi_1(G)^\top \mathbf{W}_1, \phi_2(G)^\top \mathbf{W}_2, \ldots, \phi_L(G)^\top \mathbf{W}_L],$$

with

$$\phi_l(G) = \mathbf{0} \quad \text{if} \quad l > \delta(G). \tag{13}$$

This similarity vector is then used as the graph embedding and fed into the MLP for learning. Since the inner product calculation of vectors is differentiable, the entire framework can be trained end-to-end. Under the guidance of the objective function, HSP-GKN can learn *hidden* graph features that are better aligned with downstream tasks, achieving improved performance compared to fixed similarity measurements.

### 3.5 Computational Complexity Analysis

For feature mapping, we use Dijkstra's algorithm to compute the all-pairs shortest paths for each graph, with a time complexity of $\mathcal{O}\left(n^2 \log n + nm\right)$, where $n$ is the number of vertices and $m$ is the number of edges in the graph. When generating the hierarchical feature vector representation, matrix multiplication $\mathbf{X}^\top \left(\mathbf{M}_1, \mathbf{M}_2, \ldots, \mathbf{M}_{\delta(G)}\right)$ is required, where the time complexity is $\mathcal{O}\left((\delta + 1)\, nd\right)$, where $\delta$ is the graph diameter and $d$ is the dimension of node attributes. Similarity computation uses the inner product of vectors with a time complexity of $\mathcal{O}\left((\delta + 1)\delta d\right)$. Therefore,

Table 1: Classification accuracy (± standard deviation) of our HSP, HSP-GKN, and the baselines on the TUDatasets. Best performance is highlighted in **bold**. The second-best performance is indicated with an underline. OOT means cannot completed within 24 hours. We use * to indicate the best performance among kernel methods.

| | MUTAG | NCI1 | PTC_MR | DD | BZR | PROTEINS |
|---|---|---|---|---|---|---|
| SP | $82.4 \pm 5.5$ | $72.5 \pm 2.0$ | $60.2 \pm 9.4$ | $77.9 \pm 4.5$ | $83.7 \pm 4.5$ | $74.9 \pm 3.6$ |
| WL-SP | $81.4 \pm 8.7$ | $60.8 \pm 2.4$ | $54.5 \pm 9.8$ | $76.0 \pm 3.5$ | OOT | $72.1 \pm 3.1$ |
| GH | $82.5 \pm 5.8$ | $71.0 \pm 2.3$ | $60.2 \pm 9.4$ | OOT | $82.3 \pm 7.2$ | $74.8 \pm 2.4$ |
| CORE-SP | $85.1 \pm 6.8$ | $73.8 \pm 1.4$ | $57.3 \pm 9.7$ | $79.3 \pm 3.8$ | OOT | $76.5 \pm 3.9$ |
| GraphSAGE | $83.6 \pm 9.6$ | $76.0 \pm 1.8$ | $61.7 \pm 4.9$ | $72.9 \pm 2.0$ | $81.2 \pm 4.2$ | $74.3 \pm 3.8$ |
| DiffPool | $79.8 \pm 6.7$ | $76.9 \pm 1.9$ | $61.1 \pm 5.6$ | $75.0 \pm 3.5$ | $84.5 \pm 4.2$ | $72.5 \pm 3.5$ |
| DGCNN | $84.0 \pm 7.1$ | $76.4 \pm 1.7$ | $59.5 \pm 6.9$ | $76.6 \pm 4.3$ | $81.8 \pm 4.4$ | $73.2 \pm 3.2$ |
| GIN | $84.7 \pm 6.7$ | $80.0 \pm 1.4$ | $59.1 \pm 7.0$ | $75.3 \pm 2.9$ | $85.4 \pm 5.1$ | $72.8 \pm 3.6$ |
| KerGNN | $84.2 \pm 5.1$ | $\underline{82.8 \pm 1.8}$ | $68.4 \pm 2.7$ | $\underline{78.9 \pm 3.5}$ | $81.8 \pm 3.9$ | $76.5 \pm 3.9$ |
| PathNN | $87.3 \pm 6.1$ | $81.1 \pm 1.2$ | $65.6 \pm 2.4$ | $77.0 \pm 3.1$ | $86.6 \pm 4.2$ | $75.2 \pm 3.9$ |
| MMD | $\underline{91.5 \pm 6.5}$ | $58.4 \pm 2.6$ | $62.8 \pm 1.6$ | $60.2 \pm 2.0$ | $\underline{91.0 \pm 11}$ | $77.6 \pm 2.5$ |
| GPN | $79.3 \pm 5.7$ | $80.8 \pm 0.5$ | $67.7 \pm 1.2$ | $77.0 \pm 0.8$ | $84.2 \pm 1.1$ | $75.6 \pm 0.8$ |
| Cosmo'2024 [40] | $85.2 \pm 2.2$ | $71.5 \pm 1.2$ | $60.1 \pm 1.9$ | $75.3 \pm 2.7$ | $85.5 \pm 1.9$ | $75.3 \pm 1.1$ |
| DHAKR | $89.9 \pm 1.0$ | $77.3 \pm 1.3$ | $\underline{68.8 \pm 0.9}$ | $74.6 \pm 1.7$ | $87.2 \pm 2.8$ | $77.5 \pm 0.6$ |
| HSP | $85.8 \pm 6.3^*$ | $75.2 \pm 1.5^*$ | $62.4 \pm 7.2^*$ | $78.3 \pm 2.7$ | $84.8 \pm 5.1^*$ | $73.3 \pm 3.8$ |
| HSP-GKN | $\mathbf{97.5 \pm 2.2}$ | $\mathbf{83.6 \pm 1.9}$ | $\mathbf{76.3 \pm 3.4}$ | $\mathbf{84.2 \pm 2.6}$ | $\mathbf{93.4 \pm 3.6}$ | $\mathbf{79.5 \pm 1.7}$ |

| | ENZYMES | COLLAB | IMDB-B | IMDB-M | REDDIT-B | REDDIT-M-5K |
|---|---|---|---|---|---|---|
| SP | $37.3 \pm 8.7$ | $58.8 \pm 1.2$ | $58.2 \pm 4.7$ | $39.2 \pm 2.3$ | $81.7 \pm 2.5$ | $47.9 \pm 1.9$ |
| WL-SP | $27.3 \pm 7.4$ | $58.8 \pm 1.2$ | $58.2 \pm 4.7$ | $39.2 \pm 2.3$ | OOT | OOT |
| GH | $67.7 \pm 6.5$ | $60.0 \pm 1.4$ | $59.4 \pm 3.4$ | $39.5 \pm 2.6$ | OOT | OOT |
| CORE-SP | $39.5 \pm 9.3$ | OOT | $68.5 \pm 3.9$ | $49.4 \pm 0.5$ | $\underline{91.0 \pm 1.8}$ | $54.35 \pm 0.1$ |
| GraphSAGE | $58.2 \pm 6.0$ | $73.9 \pm 1.7$ | $68.8 \pm 4.5$ | $47.6 \pm 3.5$ | $84.3 \pm 1.9$ | $50.0 \pm 1.3$ |
| DiffPool | $59.5 \pm 5.6$ | $68.9 \pm 2.0$ | $68.4 \pm 3.3$ | $45.6 \pm 3.4$ | $89.1 \pm 1.6$ | $53.8 \pm 1.4$ |
| DGCNN | $38.9 \pm 5.7$ | $71.2 \pm 1.9$ | $69.2 \pm 3.0$ | $45.6 \pm 3.4$ | $87.8 \pm 2.5$ | $49.2 \pm 1.2$ |
| GIN | $59.6 \pm 4.5$ | $75.6 \pm 2.3$ | $71.2 \pm 3.9$ | $48.5 \pm 3.3$ | $89.9 \pm 1.9$ | $\mathbf{56.1 \pm 1.7}$ |
| KerGNN | $62.1 \pm 5.5$ | $75.1 \pm 2.3$ | $74.4 \pm 4.3$ | $51.6 \pm 3.1$ | $89.5 \pm 1.6$ | $52.7 \pm 1.2$ |
| PathNN | $\mathbf{73.0 \pm 5.2}$ | $76.9 \pm 3.4$ | $72.6 \pm 3.3$ | $50.8 \pm 4.5$ | $89.2 \pm 1.1$ | $53.9 \pm 1.3$ |
| MMD | $48.4 \pm 3.3$ | $57.9 \pm 1.5$ | $59.1 \pm 2.7$ | $37.2 \pm 2.1$ | $71.3 \pm 2.7$ | $37.0 \pm 1.2$ |
| GPN | $65.6 \pm 1.3$ | $\mathbf{83.6 \pm 0.5}$ | $75.1 \pm 2.2$ | $49.4 \pm 0.9$ | $82.7 \pm 1.4$ | $46.9 \pm 0.9$ |
| Cosmo'2024 [40] | $66.4 \pm 3.2$ | $73.6 \pm 1.1$ | $69.9 \pm 1.4$ | $45.6 \pm 1.2$ | $84.9 \pm 2.5$ | $42.6 \pm 1.1$ |
| DHAKR | $64.4 \pm 4.7$ | $73.8 \pm 1.7$ | $\underline{75.2 \pm 2.6}$ | $\underline{52.1 \pm 2.5}$ | $87.4 \pm 1.3$ | $49.2 \pm 1.6$ |
| HSP | $54.6 \pm 6.8$ | $75.4 \pm 2.7^*$ | $71.5 \pm 4.9^*$ | $50.7 \pm 3.1^*$ | $83.6 \pm 3.6$ | $51.3 \pm 2.2$ |
| HSP-GKN | $\underline{70.4 \pm 3.1}$ | $\underline{80.1 \pm 1.8}$ | $\mathbf{78.5 \pm 2.8}$ | $\mathbf{55.4 \pm 1.7}$ | $\mathbf{91.7 \pm 1.2}$ | $\underline{55.7 \pm 1.3}$ |

the overall time complexity for computing the similarity is $\mathcal{O}\big(n^2 \log n + nm + (\delta + 1)(\delta + n)d\big)$. Since real-world graphs are typically sparse (i.e., $m \ll n^2$) and the graph diameter $\delta$ is usually much smaller than $n$ [42], the computational cost of the HSP kernel primarily depends on the number of nodes in the graph. Furthermore, since each graph requires feature mapping computation only once, the computational complexity of average similarity calculation is significantly lower than the aforementioned complexity. Our experiments show a linear in the number of graphs and quadratic in the number of nodes.

## 4 Experiments

In this section, we first evaluate the performance of the HSP kernel and its corresponding HSP-GKN framework on graph classification datasets. Next, we study the runtime behaviour of proposed methods on synthetic graphs and real-world datasets. Finally, we conduct ablation experiments and hyperparameter analysis on HSP-GKN.

### 4.1 Graph Classification

**Datasets.** We evaluate the proposed kernel and framework on 12 datasets from different domains and of varying scales, all included in the TUDatasets collection [43]. These datasets are categorized as follows: small molecule datasets (BZR, MUTAG, NCI1, PTC_MR), bioinformatics datasets (DD,

ENZYMES, PROTEINS), and social network datasets (COLLAB, IMDB-B, IMDB-M, REDDIT-B, REDDIT-M-5K). More details about these datasets can be found in the Appendix B.

In the HSP-GKN framework, each graph is transformed into a vector of dimension $(L+1)Ld/2$ before being fed into the model for training, which is much more compact than the original graph. Although additional preprocessing time is required, the HSP avoids processing the entire graph at each iteration. Therefore, it is comparable with GNNs in terms of overall training time.

**Baselines.** We compare the proposed kernel and model against the following graph kernels based on shortest path: shortest path kernel (SP) [34], Weisfeiler-Lehman shortest path kernel (WL-SP) [31], graphHopper kernel (GH) [36] and core shortest path kernel (CORE-SP) [35]. We also compare our model against various GNNs, including (1) classical GNNs such as GraphSAGE [44], DiffPool [45], DGCNN [46] and GIN [16]; (2) state-of-the-art GNNs such as PathNN [47], MMD [41], and GPN [48]. Furthermore, we compare our method against recently GKNNs including kerGNN [38], Cosmo'2024 [40], and DHAKR [49].

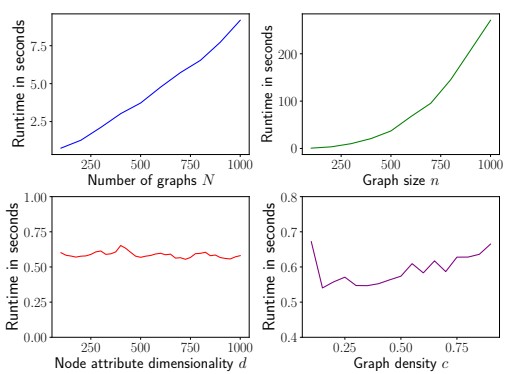

Figure 3: Dependence of runtime on $N$, $n$, $d$ and $c$ on synthetic graphs.

Table 2: Running time of different graph kernels on TUDatasets (seconds). OOT: timeout over 24 hours.

|  | ENZYMES | NCI1 | REDDIT-M-5K |
|---|---|---|---|
| SP | 7.92 | 16.33 | 12944.24 |
| WL-SP | 49.47 | 367.21 | OOT |
| GH | 356.80 | 2935.79 | OOT |
| CORE-SP | 26.48 | 49.87 | 31779.70 |
| HSP | 1.71 | 12.70 | 3869.14 |

Table 3: One-epoch running time of different GNNs on TUDatasets (seconds).

|  | ENZYMES | NCI1 | REDDIT-M-5K |
|---|---|---|---|
| GraphSAGE | 0.326 | 0.902 | 5.420 |
| DiffPool | 2.195 | 13.785 | 26.427 |
| DGCNN | 0.292 | 0.974 | 4.093 |
| GIN | 0.412 | 2.322 | 5.193 |
| HSP-GKN | 0.013 | 0.009 | 0.011 |

**Configurations.** We follow [50] and evaluate TUDatasets using a 10-fold cross-validation using their provided data splits for a fair comparison. For datasets without predefined splits, we adopt the splitting method provided in [50] and apply the same split across all baseline models. We run each experiment 10 times to obtain the average accuracy and standard deviation. For the HSP-GKN model configuration, we use a two-layer MLP, where each layer's dimension is half of the previous layer's dimension. For the selection of hyperparameters, we use Optuna [51] to tune the hyperparameters through an automated search process.

**Result analysis.** Table 1 illustrates average prediction accuracies and standard deviations. We observe that the proposed HSP-GNN framework outperforms the baselines on 9 out of the 12 datasets while it provides the second-best accuracy on the remaining 3 datasets. On the MUTAG, PTC_MR, and DD datasets, our framework achieves absolute accuracy improvements of 6.0%, 7.5%, and 5.3%, respectively, over the second-best accuracy. It is worth noting that, although we restrict the node kernel of HSP to a linear kernel, it still outperforms the shortest-path-based kernel baselines on 7 out of the 12 datasets. Furthermore, HSP-GKN achieves significantly higher classification accuracy than HSP across all datasets, demonstrating the effectiveness of our end-to-end framework. Overall, our results show that HSP-GKN achieves high performance levels on the TUDatasets.

## 4.2 Runtime Behaviour Study

Here, we experimentally examine the runtime performance of the HSP kernel and HSP-GKN framework.

**Datasets.** We assessed the behaviour on randomly generated graphs with respect to four parameters: dataset size $N$, graph size $n$, node attribute dimensionality $d$, and graph density $c$. For read-world

Table 4: Classification accuracy improvement over the HSP kernel. HSP-MLP refers to removing *hidden* graph features and directly inputting the graph feature vectors into the MLP for training.

|  | ENZYMES | NCI1 | REDDIT-M-5K | PROTEINS |
|---|---|---|---|---|
| HSP-MLP | 7.5 ↑ | 2.1 ↑ | 0.3 ↑ | 2.8 ↑ |
| HSP-GKN | 15.8 ↑ | 8.4 ↑ | 4.4 ↑ | 6.2 ↑ |

datasets, we selected 3 datasets from the TUDatasets. These datasets include large-scale dataset COLLAB, large graph dataset REDDIT-M-5K, and continuous attribute dataset ENZYMES.

**Baselines.** We compared the runtime with other graph kernel methods on real-world datasets, including SP, WL-SP, GH and CORE-SP kernel. We also evaluate the runtime in comparison to classical GNNs, including GraphSAGE, DiffPool, DGCNN and GIN.

**Configurations.** For randomly generated graphs, we kept 3 out of 4 parameters fixed at their default values and varied the fourth parameter. The default values we used were 100 for $N$, 100 for $n$, 100 for $d$, and 0.3 for the graph density $c$. All experiments were conducted on a system equipped with an 8-core Intel Xeon E5-2667v4@3.2GHz with 256 GB of RAM and an NVIDIA 3090 GPU. For the graph kernel methods used for comparison, we utilized the library provided by [52] to measure the runtime. All graph kernel methods were executed on the CPU using a single core.

**Results analysis.** On synthetic graphs, as shown in Figure 3, we observe that the HSP kernel scales linearly with dataset size $N$. When varying the number of nodes $n$ per graph, we observe that the runtime of the HSP kernel scales quadratically with $n$, as mentioned in the complexity analysis. We also observe that changing the dimension of node attributes $d$ does not significantly increase the computation time. However, when varying the graph density $c$, the computation time peaks at a density of 0.1 and 0.9. This is because, at low density, the graph diameter $\delta$ becomes large, while at high density, the number of edges $m$ is significantly higher than $n$.

With real-world datasets, as shown in Table 2, our kernel shows significantly lower computation time across all datasets, particularly on large graphs. For example, REDDIT-M-5K contains 4,999 graphs, with average node counts of 508. Some kernels even fail to complete computation within 24 hours. Since the HSP kernel is insensitive to the dimension of node attributes, it exhibits high scalability on ENZYMES datasets. In the comparison with GNNs, as shown in Table 3. Even though our method requires preprocessing, which takes slightly less time than computing the kernel matrix (see Table 2), the overall training time is significantly lower than other models on the ENZYMES and NCI1 datasets. Although preprocessing takes longer on the REDDIT-M-5K dataset due to the larger average node count, it can be parallelized to reduced preprocessing time.

## 4.3 Ablation Study

To validate the effectiveness of the proposed *hidden* graph features, we removed the *hidden* graph features and directly input the graph feature vectors into the MLP for training. As shown in Table 4, HSP-GKN achieves a more significant performance improvement over the HSP kernel. This underscores the importance of employing a trainable similarity measurement and confirms that integrating the HSP with *hidden* graph features facilitates learning task-specific graph representations that offer better performance.

## 4.4 Analysis of Hyperparameter $L$ and $N$

As can be seen in Eq. (10), HSP-GKN introduces two hyperparameters $L$ and $N$. We conduct experiments to show the effect of this parameter on four datasets. Figure 4 illustrates the performance variation of HSP-GKN when $L$ and $N$ varies. From these figures, we observe that 1) as the size of dataset increases, the performance of the method becomes less sensitive to

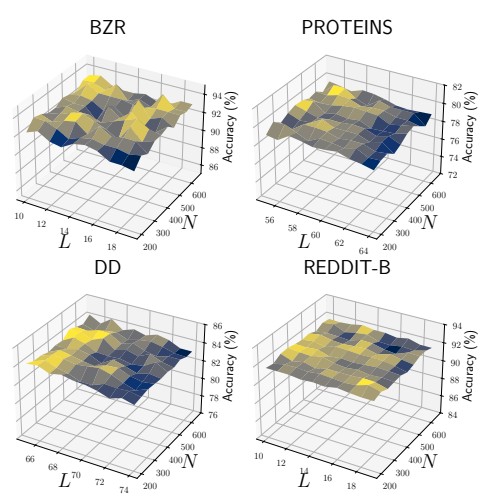

Figure 4: The sensitivity of HSP-GKN with the variation of $L$ and $N$ on four datasets.

hyperparameters; 2) the performance of the method
is overall stable across a wide range of $L$ and $N$.

## 5    Conclusion and Future Discussion

We propose the Hierarchical Shortest-Path Graph Kernel Network (HSP-GKN). In our method, we introduce the *hidden* graph features to integrate the HSP kernel similarity measurement and downstream tasks into a unified framework. Moreover, our proposed HSP kernel simultaneously preserves both the semantic and structural information of the graphs. Extensive experimental results demonstrate the effectiveness and superiority of the HSP kernel and HSP-GKN framework compared to state-of-the-art methods. In the future, we plan to extend our methods to graph regression and other tasks.

**Limitations and broader impacts**. HSP incorporates the node attribute while ignoring the edge features in graphs. We mainly focus on leveraging node features and graph topology in this work, making it broadly applicable across a wide range of datasets. However, the framework is flexible and can be extended to support edge attributes in several ways. One possible extension is concatenating the edge attributes along the shortest path with the corresponding node attributes into a unified vector representation. Since the shortest paths are already computed as part of the kernel, this concatenation would introduce minimal additional computational overhead. Alternatively, we can define a separate kernel for edge features to measure the similarity between two shortest paths based on their edge attributes. This edge-based similarity can then be combined with node-based similarity to provide a more comprehensive comparison between shortest paths. We leave this as a promising direction for future work and believe that such an extension can further enhance the model's expressiveness on attributed graphs with rich edge information. The end-to-end manner in graph kernel methods is inspiring for the community.

## Acknowledgements

This work was supported by the National Natural Science Foundation of China (NSFC) (Grant No. 62506102, 62562026), the Key Research and Development Program of Hainan Province (Grant No. ZDYF2024GXJS014, ZDYF2023GXJS163), and the Natural Science Foundation of Hainan University (No. XJ2400009401).

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

# A    Proof of the Statements in Section 3.2

## A.1    Derivation of Eq. (4)

Given two shortest path $\pi_l = [v_1, v_2, \ldots, v_{l+1}]$ and $\pi'_l = [v'_1, v'_2, \ldots, v'_{l+1}]$. As mentioned in preliminary work, we have $a(\pi) = \text{concat}(a(v_1), a(v_2), \ldots, a(v_{l+1}))$. From Eq. (2), we know

$$k_{path}(\pi, \pi') = \sum_{i=1}^{|\pi|+1} \langle a(v_i), a(v'_i) \rangle. \tag{14}$$

Now we have

$$k_{path}(\pi, \pi') = \langle a(\pi), a(\pi') \rangle. \tag{15}$$

By substituting into Eq. (1), we can get

$$
\begin{aligned}
k_{\text{hsp}}(G, G') &= \sum_{l=1}^{L} \left[ \sum_{\pi \in \mathcal{SP}_l(G)} \sum_{\pi' \in \mathcal{SP}(G')} \langle a(\pi), a(\pi') \rangle \right] \\
&= \sum_{l=1}^{L} \left\langle \sum_{\pi \in \mathcal{SP}_l(G)} a(\pi), \sum_{\pi' \in \mathcal{SP}(G')} a(\pi') \right\rangle \\
&= \sum_{l=1}^{L} \langle \phi_l(G), \phi_l(G') \rangle. 
\end{aligned}
\tag{16}
$$

Eq. (4) has been derived.

## A.2    Proof of Lemma 3.1

**Definition A.1.** (Positive definite kernel) Let $k : \mathcal{X}$ be a set. $k : \mathcal{X} \times \mathcal{X} \to \mathbb{R}$ is a positive definite kernel if $k(x, y) = k(y, x)$ and for any $n \in \mathbb{N}$, $x_1, \ldots, x_n \in \mathcal{X}$ and $c_1, \ldots, c_n \in \mathbb{R}$

$$\sum_{i=1}^{n} \sum_{j=1}^{n} c_i c_j k(x_i, x_j) \geq 0. \tag{17}$$

In Eq. (9), we mentioned that

$$k(G_i, G_j) = \boldsymbol{\Phi}(G_i) \odot \boldsymbol{\Phi}(G_j).$$

Now we zero-extend $\boldsymbol{\Phi}(G_j)$ and $\boldsymbol{\Phi}(G_j)$ to a sufficiently large dimension $D$ that exceeds the dimension of any possible graph, that is, $\boldsymbol{\Phi}_{pad}(G) = [\boldsymbol{\Phi}(G), 0] \in \mathbb{R}^D$, this operation does not change the process of feature mapping. Now we have

$$k(G_i, G_j) = \langle \boldsymbol{\Phi}_{pad}(G_i), \boldsymbol{\Phi}_{pad}(G_j) \rangle. \tag{18}$$

First, $k(G_i, G_j)$ is obviously symmetric. Given any $n \in \mathbb{N}$, non-empty set of graphs $\{G_1, G_2, \ldots, G_n\}$ and $c_1, \ldots, c_n \in \mathbb{R}$. We first define the quadratic form:

$$Q = \sum_{i=1}^{n} \sum_{j=1}^{n} c_i c_j k(G_i, G_j).$$

With refernce to Eq. (18), we have:

$$
\begin{aligned}
Q &= \sum_{i=1}^{n} \sum_{j=1}^{n} c_i c_j \langle \boldsymbol{\Phi}_{\text{pad}}(G_i), \boldsymbol{\Phi}_{\text{pad}}(G_j) \rangle \\
&= \left\langle \sum_{i=1}^{n} c_i \boldsymbol{\Phi}_{\text{pad}}(G_i), \sum_{j=1}^{n} c_j \boldsymbol{\Phi}_{\text{pad}}(G_j) \right\rangle \\
&= \left\langle \sum_{i=1}^{n} c_i \boldsymbol{\Phi}_{\text{pad}}(G_i), \sum_{i=1}^{n} c_i \boldsymbol{\Phi}_{\text{pad}}(G_i) \right\rangle.
\end{aligned}
\tag{19}
$$

Table 5: Dataset statistics and properties of TUDatasets.

| Dataset | Properties | | | | | |
|---|---|---|---|---|---|---|
| | Number of graphs | Number of classes | Avg number of vertices | Avg number of edges | Vertex labels | Continuous vertex attributes |
| MUTAG | 188 | 2 | 17.93 | 19.79 | ✓ | × |
| NCI1 | 4110 | 2 | 29.87 | 32.30 | ✓ | × |
| PTC_MR | 344 | 2 | 14.29 | 14.69 | ✓ | × |
| DD | 1178 | 2 | 284.32 | 715.66 | ✓ | × |
| BZR | 405 | 2 | 35.75 | 38.36 | ✓ | ✓ |
| PROTEINS | 1113 | 2 | 39.06 | 72.82 | ✓ | ✓ |
| ENZYMES | 600 | 6 | 32.6 | 62.14 | ✓ | ✓ |
| COLLAB | 5000 | 3 | 74.49 | 2457.78 | × | × |
| IMDB-BINARY | 1000 | 2 | 19.77 | 96.53 | × | × |
| IMDB-MULTI | 1500 | 3 | 13.00 | 65.94 | × | × |
| REDDIT-BINARY | 2000 | 2 | 429.63 | 497.75 | × | × |
| REDDIT-MULTI-5K | 4999 | 5 | 508.52 | 594.87 | × | × |

Let $v = \sum_{i=1}^{n} c_i \mathbf{\Phi}_{pad}(G_i)$, now we have $Q = \langle v, v \rangle = \|v\|^2$. Clearly, $Q \geq 0$, therefore $k(G_i, G_j)$ is positive definite.

### A.3 Proof of Lemma 3.2

Assume there is a shortest-path of length $l$ from vertex $v_1$ to $v_{l+1}$, denoted as $\pi_l = [v_1, \ldots, v_l, v_{l+1}]$. Then, there exists a path of length $l-1$ from vertex $v_1$ to $v_l$, $\pi_{l-1} = [v_1, v_2, \ldots, v_l]$. If $\pi_{l-1}$ is not the shortest-path in graph $G$, meaning there is a shorter shortest-path from $v_1$ to $v_l$, then there must exist a shortest-path from $v_1$ to $v_{l+1}$ with length less than $l$, which contradicts the assumption. Therefore, $\pi_{l-1}$ is the shortest-path of length $l-1$ from vertex $v_1$ to $v_l$. The proof for other lengths is similar.

## B Dataset statistics.

We evaluated the proposed methods on 12 publicly available graph classification datasets including 4 small molecule datasets: BZR, MUTAG, NCI1 and PTC_MR, 3 bioinformatics datasets: DD, ENZYMES and PROTEINS, as well as 6 social network datasets: COLLAB, IMDB-B, IMDB-M, REDDIT-B, REDDIT-M-5K).

A summary of the 12 datasets is given in Table 5.

