# OpenReview forum: "Hierarchical Shortest-Path Graph Kernel Network"
_NeurIPS.cc/2025/Conference — NeurIPS 2025 spotlight_

### Official Review · Reviewer_3EPr · 2025-06-21

**Clarity:** 3
**Significance:** 3
**Originality:** 3
**Rating:** 5
**Confidence:** 4

**Summary:**

This paper proposes a graph learning framework, HSP-GKN. The framework first introduces a shortest-path-based graph kernel that captures graph structure and semantics via an explicit feature mapping, which aggregates all shortest paths of the same length. Subsequently, the authors embed this HSP kernel into an end-to-end network. The network generates graph embeddings for downstream tasks by computing the similarity between the input graph's HSP features and a set of learnable hidden graph features. This design allows the graph representation to be optimized in a task-driven manner.

**Questions:**

See above.

Additionally, I am curious about the performance of HSP-GKN on other tasks, such as graph regression.

**Ethical Concerns:**

["NO or VERY MINOR ethics concerns only"]

**Final Justification:**

I have checked the comments from the other reviewers and the corresponding replies from the author. Regarding the novelty concern, I believe the major contribution of this paper lies in the computationally efficient shortest-path kernel and the end-to-end optimization framework. I confirm my recommendation to accept the paper.

**Limitations:**

Yes.

**Paper Formatting Concerns:**

N/A.

**Quality:**

4

**Strengths And Weaknesses:**

**Strengths**

1. The paper is well-written and clearly structured. The motivation, methodology, and experiments are presented logically.

2. The introduction of learnable hidden graph features that are matched against the graph's HSP features is an interesting design. Introducing flexibility into a traditionally fixed kernel approach.

3. The experimental evaluation is quite thorough, covering 12 public datasets from various domains.

**Weaknesses**

1. The term "Hierarchical" may be imprecise. The method groups features by path length rather than constructing a true multi-scale or coarse-to-fine representation.

2. According to my understanding, there should be a missing symbol $l$ in Eq. (1). Additionally, certain equations, such as Eq. (10), would benefit from more detailed explanations to enhance clarity.

3. Loss of structural information. By collapsing all same-length shortest paths into one vector, the model might not distinguish between graphs with different path topologies but similar aggregated node features.

4. In the ablation study, more detailed explanations are needed to clarify the distinctions between HSP-MLP and HSP-GKN. Given their seemingly similar architectures, it would be helpful for the authors to elaborate on the specific design differences and provide insights into why HSP-GKN yields superior performance.

---

> ### Author Rebuttal · Authors · 2025-07-28
>
> **Weaknesses 1:** We appreciate the reviewer’s insightful comment regarding the use of the term “Hierarchical”. While it is true that our method does not construct a multi-scale or coarse-to-fine representation, we use the term to emphasize the multi-level structural abstraction captured by varying shortest-path lengths. Each path length corresponds to a distinct level of structural granularity: short paths capture local neighborhood patterns, whereas longer paths reflect broader graph topology. We acknowledge that “multi-level” or “multi-scale path-based representation” might be alternative terms, and we are open to refining the terminology in the revised version to improve clarity.
>
> **Weaknesses 2:** Thanks for the comment. We will check the manuscript’s content, correct errors as per your comments, and carefully double-check the writing. The expression in Eq. (10) defines the set of hidden graph features as a collection of learnable, hierarchical representations. Specifically, $\phi_{l}^{(i)}(\mathcal{G})$ denotes the $i$-th feature corresponding to paths of length $l$, and $n_l$ indicates the number of such features at level $l$. These features capture structural patterns at different levels of granularity, enabling the model to learn multi-scale representations of the graph.
>
> **Weaknesses 3:** We thank the reviewer for pointing out the concern regarding potential loss of structural information when aggregating same-length paths. We would like to clarify that our similarity vector construction is fundamentally different from naive feature collapsing. While typical GNN pooling compresses node representations using operations such as max or mean—often at the risk of discarding fine-grained structural distinctions—our method explicitly preserves multi-level structural information by aggregating node features within shortest paths of each specific length. These aggregated features are then compared with trainable anchors to form a similarity vector, where each dimension corresponds to a distinct level of structural abstraction. This design allows the model to retain path-level topological characteristics, offering a more structured and semantically aligned representation than standard pooling techniques.
>
>
> **Weaknesses 4:** Thanks for the comment. For HSP-MLP, let $\Phi(G)$ denote the vectorial representation of a graph, defined as $\Phi(G) = $CONCAT$(\phi_1(G), \ldots, \phi_L(G))$. This representation is directly fed into an MLP for training. However, the hierarchical characteristics of the graph are not explicitly considered in this process. In contrast, HSP-GKN computes a similarity vector by measuring the similarity between the learnable anchor features and the hierarchical features $\phi_l(G)$ of a given graph. In this case, each element of the similarity vector corresponds to a different level of hierarchy. This vector is then used as the input to the MLP for downstream training.
>
> **Other comment:** **Other tasks** Thanks for the comment. Unlike the graph classification task, most widely used datasets in the graph classification task do not have edge attributes. However, in graph regression datasets such as ZINC and QM9, which have discrete or continuous edge attributes. The main reason why there is no regression task is the lack of full utilization of edge information in the graphs. Furthermore, the prediction target in graph regression tasks lies in a continuous space, which may require designing models that are more sensitive to the predicted values. We agree that extending this approach to the regression task is a promising direction, and we plan to explore it in future work.

---

### Official Review · Reviewer_wTHL · 2025-06-21

**Clarity:** 3
**Significance:** 2
**Originality:** 2
**Rating:** 3
**Confidence:** 4

**Summary:**

This paper proposes the Hierarchical Shortest-Path Graph Kernel (HSP) and a corresponding Graph Kernel Network (HSP-GKN) framework, aiming to combine the benefits of graph kernels and neural networks. The HSP kernel encodes graph structure and semantics through shortest-path-based hierarchical feature mappings. HSP-GKN further introduces trainable hidden graph features to produce similarity vectors for downstream learning. The method is theoretically justified and evaluated on 12 graph classification benchmarks, showing favorable performance compared to classical kernels and GNNs.

**Questions:**

1. The integration of shortest-path kernels and neural networks seems incremental. Can the authors clarify what is technically novel compared to prior works using similar ideas [1]?

2. The method is only tested on small datasets. Can the authors provide results on large-scale benchmarks to support scalability claims?

3. The method does not support edge features. Is it possible to extend HSP-GKN to incorporate edge attributes?

4. Performance trends in Figure 4 are hard to read. Would the authors consider using a heatmap with a color bar to better illustrate sensitivity to L and N?

[1] DHAKR: Learning Deep Hierarchical Attention-Based Kernelized Representations for Graph Classification. AAAI 2025

**Ethical Concerns:**

["NO or VERY MINOR ethics concerns only"]

**Final Justification:**

The rebuttal clarifies technical differences from prior work and outlines potential extensions. However, the core contribution remains incremental, and the lack of edge attribute support limits practical scope such as molecular property prediction. These concerns are not fully addressed in the current version. I therefore maintain my original scores.

**Limitations:**

The limitations section correctly acknowledges the exclusion of edge features. However, implications of this on domains like molecular property prediction are not deeply discussed.

**Quality:**

2

**Strengths And Weaknesses:**

Strengths

1. This paper studies an important problem of  integrating graph kernels and neural networks.

2. This paper is well organized. The technical details are also easy to follow.

Weaknesses
1. The novelty of the proposed framework in this paper is incremental. Integrating shortest-path kernels and neural networks hierarchically is not a novel idea, which has been studied in prior works [1].

2. Datasets used in the paper are relatively small. To convincingly demonstrate scalability, it would be valuable to evaluate HSP-GKN on large-scale graph datasets.

3. The method does not support edge attributes, which are vital in many practical graphs (e.g., molecular graphs).

4. In Figure 4, it's difficult to discern the exact performance trends. Using a heatmap with a color bar could improve readability.

[1] DHAKR: Learning Deep Hierarchical Attention-Based Kernelized Representations for Graph Classification. AAAI 2025

---

> ### Author Rebuttal · Authors · 2025-07-28
>
> **Question 1:** **Technical contribution:** We appreciate the reviewer’s concern regarding the novelty of our approach relative to existing methods such as DHAKR [1]. While both HSP-GKN and DHAKR aim to integrate shortest-path kernels with deep architectures, there are significant differences which we summarize below: **(1) Substructure Encoding: Count-based vs. Similarity-based:** DHAKR encodes substructures by explicitly counting occurrences of patterns (e.g., shortest path length or WL subtrees), resulting in discrete, sparse frequency vectors. Moreover, these counts are based purely on structure and do not incorporate node attributes. This approach inevitably leads to the loss of node semantics, thereby negatively impacting the model's performance. In contrast, HSP-GKN encodes topological information implicitly by introducing hierarchical feature mapping that maps graphs into vector representations. These vector representations are then used to measure similarity with learnable graph anchors. **(2) End-to-End Learning Objective: Attention weighting vs. Representation learning:** DHAKR aims to learn attention weights over predefined substructures, identifying which substructure counts are more relevant to the task. However, this approach inherently relies on a fixed set of symbolic patterns, which may limit its capacity to generalize beyond the predefined structures. In contrast, HSP-GKN is designed to learn hidden graph representations in a kernel-induced space directly. Through hierarchical feature mapping and anchor-based similarity computation, the model learns a task-adaptive, continuous graph embedding that implicitly captures structural and attribute-level semantics, rather than weighting fixed symbolic patterns. This enables more flexible and expressive representations, without being constrained by manually selected substructures. **(3) Computational Complexity:**            Specifically, the computational cost of DHAKR is $\mathcal{O}(N^2)$ in the number of graphs $N$ and $\mathcal{O}(n^3)$ in the number of nodes per graph $n$. In contrast, HSP-GKN scales more efficiently with $\mathcal{O}(N)$ and $\mathcal{O}(n^2)$, respectively. From a model complexity perspective, HSP-GKN introduces a significantly more compact and scalable framework, DHAKR constructs multiple kernel matrices across layers and aggregates them via attention mechanisms. Therefore, HSP-GKN has fewer parameters than attention-based aggregation in DHAKR, and achieves superior scalability with linear complexity in the number of graphs.
>
> In summary, HSP-GKN differs from DHAKR in three fundamental ways: it learns similarity-based representations rather than relying on lossy count-based encodings, allowing it to retain both structural and attribute information; it operates in a continuous embedding space instead of weighting fixed symbolic patterns, enabling greater expressiveness and generalization; and it offers improved scalability with lower computational complexity and fewer model parameters. We will add more discussion in the related work in the revision.
>
> **Question 2:** Thanks for the comment. We have added OGBG-molhiv and REDDIT-M-12K datasets. The summary of the dataset is shown in Table 1. The performance comparison is shown in Table 3,4, the runtime comparison is shown in Table 2. The results show that our method consistently outperforms existing approaches across all datasets, demonstrating the effectiveness of our proposed framework against its competitors. The results will be added to the revision.
>
> ## Table 1 Dataset statistics
> |              | Number of graphs | Avg number of vertices | Avg number of edges |
> | ------------ | :--------------: | :--------------------: | :-----------------: |
> | OGBG-molhiv  |      41127       |         25           |         27          |
> | REDDIT-M-12K |      11929       |        391           |         456         |
>
> ## Table 2 Runtime (seconds)
> |              | SP    | CORE-SP | WL-SP | GH   | Ours |
> | ------------ | ----- | ------- | ----- | ---- | ---- |
> | REDDIT-M-12K | 22777 | OOT     | OOT   | OOT  | 6845 |
>
> ## Table 3 Performance Comparison
> |  | **SP** |**GraphSAGE** |**DiffPool** |**DGCNN** |**GIN**|**RWNN**|**KerGNN**|**PathNN**|**MMD**|**GPN** |**HSP** |**HSP-GKN** |
> | --- | --- |---|---|---|---|---|---|---|---|---|---|---|
> | **REDDIT-M-12K** | 35.79±1.9 | 43.5 ± 1.0 | 44.4±1.4 | 43.9±1.8 | `46.7±1.6` | 45.2±1.1 |43.8±1.0 |45.9±1.5 |43.1±1.7 |45.7±1.2 |39.2±1.5|**47.8±1.7**|
>
>  ## Table 4 Performance Comparison
> |  | **GIN** | **GCN** | **pathNN** |**AgentNet[2]** |**HSP-GKN** |
> | --- | --- | --- | ---|---|---|
> | **OGBG-molhiv** | 75.6±1.4 | 76.0±0.9 | 79.2±1.9 | 78.3±0.7 | **79.7±1.3** |
>
> **Question 3:** We thank the reviewer for pointing this out. Indeed, the current version of HSP-GKN focuses on node attributes and graph topology. Most commonly used graph classification benchmarks do not contain edge attributes. Among the 12 datasets in this paper, only 2 provide edge attribute information; the other 10 datasets do not include edge attributes. However, the framework is flexible and can be extended to support edge attributes in several ways.
>
> One possible extension is concatenating the edge attributes along the shortest path with the corresponding node attributes into a unified vector representation. Since the shortest paths are already computed as part of the kernel, this concatenation would introduce minimal additional computational overhead. Alternatively, we can define a separate kernel for edge features to measure the similarity between two shortest paths based on their edge attributes. This edge-based similarity can then be combined with node-based similarity to provide a more comprehensive comparison between shortest paths. We leave this as a promising direction for future work and believe that such an extension can further enhance the model’s expressiveness on attributed graphs with rich edge information.
>
> **Question 4:** Thanks for the comment. Due to the rebuttal policy limits, we are unable to submit figures. Following your suggestions, we will add a heatmap to improve readability in the revised manuscript.
>
> **Limitations:** Thanks for your suggestion about the limitations section. We appreciate the reviewer’s observation that while we have acknowledged the exclusion of edge features, the implications of this limitation on specific domains, such as molecular property prediction, have not been deeply discussed. We agree that this is an important point, as edge features (e.g., bond types or interaction strengths in molecular graphs) can significantly influence model performance in such applications. We will also discuss in the revision how to extend the method to graphs with edge attributes, as discussed in our response to Question 3. We hope this addition provides a clearer understanding of the implications.
>
> [1] DHAKR: Learning Deep Hierarchical Attention-Based Kernelized Representations for Graph Classification. AAAI 2025
>
> [2] Agent-based graph neural networks. ICLR, 2023.

---

> > ### Comment · Reviewer_wTHL · 2025-08-05
> >
> > Thank you for your detailed rebuttal. Your clarifications on technical differences and possible extensions are appreciated. However, I find the contribution still somewhat incremental, and the lack of edge attribute support remains a limitation, which limits practical scope such as molecular property prediction. I will maintain my original scores.

---

> > > ### Author Response · Authors · 2025-08-05
> > >
> > > Dear Reviewer wTHL,
> > >
> > > Thanks for your appreciation of the technical distinctions and possible extensions highlighted in our rebuttal. Regarding the support for edge attributes, we believe it is a critical yet under-explored research topic in the domain of graph kernel learning. However, this study primarily focuses on addressing the challenges of end-to-end learnability and computational efficiency in graph kernels.
> > >
> > > Inspired by your comments, we plan to investigate the incorporation of edge attributes as part of our future work. Thank you once again for your insightful and constructive feedback!
> > >
> > > Best regards,
> > >
> > > Authors of Paper 8752

---

> ### Author Response · Authors · 2025-08-05
>
> Dear Reviewer wTHL,
>
> The Author-Reviewer discussion phase is now halfway. All concerns are replied to one by one in the rebuttal, and we hope the response will meet your approval. Are there other concerns you want to discuss with us? Feel free to leave your message, thus we can address your comments in time during the discussion phase.
>
> Best regards,
>
> Authors of Paper 8752

---

### Official Review · Reviewer_QL9x · 2025-06-25

**Clarity:** 3
**Significance:** 3
**Originality:** 3
**Rating:** 5
**Confidence:** 2

**Summary:**

The authors propose a new method called hierarchical shortest-path graph kernel network. First they propose the hierarchical shortest path kernel, a kernel based on the graphHopper kernel, that can be computed efficiently. They then use the similarity of the resulting feature vectors to a hidden (learnable) graph to learn solving downstream tasks. Experimental results show that the proposed kernel is efficiently computable also in practice and the framework reaches the highest accuracy in almost all cases (and the second-best in the others).

**Questions:**

**Q1** Why do you not compare to [26] and [27] (references from paper) in the experimental evaluation?

**Q2** Why are there no regression tasks in the experiments?

**Q3** Why are edge labels not considered? And for the experiments: Were the edge labels removed in preprocessing or did some methods use them (or do the datasets not have edge labels)?

In order to improve the paper, please consider comparing also to the more recent graph kernels mentioned above.

**Ethical Concerns:**

["NO or VERY MINOR ethics concerns only"]

**Final Justification:**

The authors have addressed my concerns and provided additional results. Having also read the other reviews, I tend more towards acceptance.

**Limitations:**

The limitation section is a bit short. Limitations should not only be stated, but also discussed.

**Paper Formatting Concerns:**

No concerns.

**Quality:**

2

**Strengths And Weaknesses:**

I really like the idea and think that combining graph kernels with neural methods is an important research direction. The experimental results are very good; the proposed method improves accuracy on most datasets. However, the datasets used are rather old and outdated. The authors should consider running experiments on more recent datasets.
For the graph kernels, there are newer (and/or better) methods available, for example the WLOA [1], WWL [2] or GWL [3] kernel. The proposed method should also be compared to them.

[1] On Valid Optimal Assignment Kernels and Applications to Graph Classification, Kriege et al., 2016

[2] Wasserstein Weisfeiler-Lehman Graph Kernels, Togninalli et al., 2022

[3] A generalized Weisfeiler-Lehman graph kernel, Schulz et al., 2022


# Other comments

It might be a good idea to use a shorter acronym for the method.

The font size in Figure 1 is too small.

l121 *a* is missing in the definition of G=(V,E)

l137 How is $k_{path}$ defined in the original graphHopper kernel?

Figure 2: Similartiry -> Similarity

l208 . missing

Figure 3: font size is too small.

l294 Please explain what read-word datasets are (or change to real-world if that is just a typo).

l306ff/Figure3: The plots regarding N and n, look the same, so it is not clear how the desscription fits them. Please make all plots have the same axis labels (runtime instead of sqrt of runtime for the one with n), then it can be seen with one glance.

l344 "Hierarchical" missing before Shortest-Path

l351 Please rephrase the last sentence.

---

> ### Author Rebuttal · Authors · 2025-07-28
>
> **Question 1:** Thank you for the suggestion. We have conducted experiments with the suggested method in Table 1 and Table 2, and will include the results in the experiments section in the revised version. Experimental results show that our method achieves outstanding performance, even when compared to more advanced state-of-the-art approaches. We appreciate the insightful comment and believe that including this comparison will further strengthen our experimental analysis.
>
> ## Table 1 Performance Comparison
> |          |  MUTAG  |   PTC-MR   |    NCI1    | PROTEINS | IMDB-B |   IMDB-M    |
> | :------: | :--------: | :--------: | :--------: | :---------: | :--------: | :---------: |
> | GNN-LoFI[4] | 86.7±2.0 | 65.7±5.2 | 79.4±0.4 | 75.1±1.0  | 76.5±2.1 | 48.8±1.2  |
> | Baseline[5] | 85.2±2.2 | 60.1±1.9 | 71.5±1.2 | 75.3±1.1  | 69.9±1.4 | 45.6±1.2 |
> | HSP-GKN  |  **97.5±2.2**  | **76.3±3.4** |  **83.6±1.9**  | **79.5±1.7** |     **78.5±2.8**     | **78.5±2.8** |
>
> ## Table 2 Newer graph kernels
> |         |  MUTAG  |   PTC-MR   |    NCI1    | PROTEINS |    D&D     |  ENZYMES   |   IMDB-B    |     BZR     |
> | :-----: | :--------: | :---------: | :--------: | :---------: | :--------: | :--------: | :---------: | :---------: |
> |  WL-OA[1]  |  84.5±1.7  |  63.6±1.5   |  **86.1±0.2**  |      76.4±0.4      |  79.2±0.4  |  59.9±1.1  |      72.6±5.5      |      86.5±0.5      |
> |   WWL[2]   | 87.2±1.5 | 66.3±1.2 | 85.7±0.2 | 74.2±0.5  | 79.6±0.5 | 59.1±0.8 | 74.3±0.8 | 84.4±2.0 |
> | R-WL[6] | 89.5±5.3 | 57.0±2.6 | 80.4±0.8 | 75.1±4.5 | OOT | 49.2±5.9 | 71.9±2.4 | 87.8±2.6 |
> | HSP-GKN |  **97.5±2.2**  |      **76.3±3.4**      |  83.6±1.9  | **79.5±1.7** |     **84.2±2.6**     |  **70.4±3.1**  | **78.5±2.8** | **93.4±3.6** |
>
> **Question 2:** Thanks for the comment. Unlike the graph classification task, most widely used datasets in the graph classification task do not have edge attributes. However, in graph regression datasets such as ZINC and QM9, which have discrete or continuous edge attributes. The main reason why there is no regression task is the lack of full utilization of edge information in the graphs. Furthermore, the prediction target in graph regression tasks lies in a continuous space, which may require designing models that are more sensitive to the predicted values. We agree that extending this approach to the regression task is a promising direction, and we plan to explore it in future work.
>
> **Question 3:** Thanks for the comment. Most commonly used graph classification benchmarks do not contain edge attributes, which limits the feasibility of incorporating them into a kernel design that aims to be generally applicable. Among the 12 datasets in this paper, only 2 provide edge attribute information; the other 10 datasets do not include edge attributes. In addition, most existing graph kernels—such as SP, WL-SP, GH, and CORE-SP—are primarily designed to capture structural patterns and generally disregard edge attributes. Therefore, we mainly focus on leveraging node features and graph topology in this work, making it broadly applicable across a wide range of datasets. However, our method can be extended to incorporate edge attributes when available. In future work, we plan to explore variants of the proposed kernel that can explicitly encode edge information for tasks or datasets where edge features play a crucial role.
>
> **Other comments:** We sincerely appreciate your thorough review. 1. There are indeed some typos—for example, “read-word” should be “real-world”. We will check the manuscript’s content, correct errors as per your comments, and carefully double-check the writing. 2. **The definition of $k_{path}$** in the original graphHopper (GH) kernel is the sum of node kernels. The difference is the choice of the node kernel $k_{node}$: while the GH kernel allows for arbitrary kernels, our proposed HSP kernel restricts $k_{node}$ to a linear kernel. This choice enables a fundamentally different and more efficient computation strategy, significantly reducing kernel computation time, as shown in Table 2 of the paper. Furthermore, using a linear kernel allows for an explicit mapping of graphs into a vector space, which serves as the foundation for the subsequent end-to-end HSP-GKN framework. 3. **Figure3** was intended to illustrate that the computation time grows quadratically with the number of nodes in the graph. Following your suggestions, we will make all plots have the same axis labels in the revision.
>
> **Limitations:** Thanks for the comment. Following your suggestions, we will add more discussion about the limitations of the exclusion of edge features in the final paper, including but not limited to how our method can be extended to handle graphs with edge attributes, as well as the importance of edge information in specific domains such as molecular property prediction.
>
> [1] On Valid Optimal Assignment Kernels and Applications to Graph Classification, Kriege et al., 2016
>
> [2] Wasserstein Weisfeiler-Lehman Graph Kernels, Togninalli et al., 2022
>
> [3] A generalized Weisfeiler-Lehman graph kernel, Schulz et al., 2022
>
> [4] Gnn-lofi: a novel graph neural network through localized feature-based histogram intersection. 2024
>
> [5] Graph kernel neural networks. 2024
>
> [6] A generalized Weisfeiler-Lehman graph kernel, Schulz et al., 2022

---

> > ### Comment · Reviewer_QL9x · 2025-08-04
> > **Reply to Rebuttal**
> >
> > Thank you for addressing my concerns and also providing additional results!
> > Having read also the other reviews and the corresponding rebuttals, I updated my score accordingly.

---

> > > ### Author Response · Authors · 2025-08-05
> > >
> > > Dear Reviewer QL9x,
> > >
> > > We sincerely thank you for the detailed and valuable comments on our paper.
> > >
> > > Best regards,
> > >
> > > Authors of Paper 8752

---

### Official Review · Reviewer_y3Cv · 2025-06-30

**Clarity:** 3
**Significance:** 3
**Originality:** 3
**Rating:** 5
**Confidence:** 5

**Summary:**

This paper presents the Hierarchical Shortest-Path Graph Kernel Network (HSP-GKN), which seamlessly combines graph kernel similarity estimation with neural network optimization in an end-to-end fashion. The key idea is that the authors designed the hierarchical feature mapping by degenerating the GH kernel. This feature mapping decreases the computational time and renders its combination with neural networks more straightforward. Based on this kernel, the paper proposes the concept of hidden graph features, which empowers neural networks to capture discriminative representations for downstream tasks. The authors further present comprehensive experimental analyses and results that demonstrate the promising performance of HSP-GKN across diverse benchmarks.

**Questions:**

1. How does HSP-GKN perform on other large-scale datasets (e.g., OGB)?
2. How do the hidden graph features contribute to improved model performance? It seems that it merely transforms the graph into a vector.
3. HSP only compares paths of the same length. Compared to the SP kernel, does this reduce the expressive power of the kernel?

**Ethical Concerns:**

["NO or VERY MINOR ethics concerns only"]

**Final Justification:**

I appreciate the authors' thoughtful rebuttal. Their responses have addressed my concerns. Accordingly, I recommend accepting the paper.

**Limitations:**

Yes

**Quality:**

3

**Strengths And Weaknesses:**

Strengths
1. This paper challenges conventional graph kernel methodologies by examining the importance of graph-to-vector conversion from a feature mapping perspective. The proposed approach is simple yet offers meaningful insights.
2. The concept of HSP-GKN is both novel and well-justified. The authors offer a thorough analysis highlighting the need to unify kernel-based similarity measurement with objective optimization within a single integrated framework.
3. The graph classification and runtime results of the proposed HSP and HSP-GKN are convincing, demonstrating their strong performance and efficiency.

Weaknesses
1. A minor shortcoming is the absence of a detailed discussion on how the proposed HSP kernel differs from prior approaches, such as the Core-SP kernel. The authors are encouraged to discuss the difference between the HSP and prior kernel methods.
2. To reduce computational complexity, HSP restricts the node kernel to a linear kernel. It would be better to discuss the potential drawbacks of this design choice.
3. It is recommended that the authors provide a more detailed description of the datasets in Appendix B to enhance the clarity and reproducibility of the experiments.
4. An in-depth analysis of how the learned hidden graph features contribute to the improved model performance gains would strengthen the paper.
5. In the related work section, the authors should further clarify the differences between the proposed approach and existing GKNNs methods, such as RWNN [1] and KerGNN [2].
6. The authors should analyze the expressiveness of the HSP kernel since HSP only compares paths of the same length, which can be done in the revision.

[1] Random walk graph neural networks. NeurIPS, 2020.

[2] Kergnns: Interpretable graph neural networks with graph kernels. AAAI, 2022.

---

> ### Author Rebuttal · Authors · 2025-07-28
>
> **Question 1:** Thanks for the comment. We have added OGBG-molhiv and REDDIT-M-12K datasets. The summary of dataset is shown in Table 1. The performance comparison is shown in Table 3,4, the runtime comparison is shown in Table 2. Experimental results show that our method achieves outstanding performance. The results will be added to the revision.
>
> ## Table 1 Dataset statistics
> |              | Number of graphs | Avg number of vertices | Avg number of edges |
> | ------------ | :--------------: | :--------------------: | :-----------------: |
> | OGBG-molhiv  |      41127       |         25           |         27          |
> | REDDIT-M-12K |      11929       |        391           |         456         |
>
> ## Table 2 Runtime (seconds)
> |              | SP    | CORE-SP | WL-SP | GH   | Ours |
> | ------------ | ----- | ------- | ----- | ---- | ---- |
> | REDDIT-M-12K | 22777 | OOT     | OOT   | OOT  | 6845 |
>
> ## Table 3 Performance Comparison
> |  | **SP** |**GraphSAGE** |**DiffPool** |**DGCNN** |**GIN**|**RWNN**|**KerGNN**|**PathNN**|**MMD**|**GPN** |**HSP** |**HSP-GKN** |
> | --- | --- |---|---|---|---|---|---|---|---|---|---|---|
> | **REDDIT-M-12K** | 35.79±1.9 | 43.5 ± 1.0 | 44.4±1.4 | 43.9±1.8 | `46.7±1.6` | 45.2±1.1 |43.8±1.0 |45.9±1.5 |43.1±1.7 |45.7±1.2 |39.2±1.5|**47.8±1.7**|
>
>  ## Table 4 Performance Comparison
> |  | **GIN** | **GCN** | **pathNN** |**AgentNet[1]** |**HSP-GKN** |
> | --- | --- | --- | ---|---|---|
> | **OGBG-molhiv** | 75.6±1.4 | 76.0±0.9 | 79.2±1.9 | 78.3±0.7 | **79.7±1.3** |
>
> **Question 2:** Thanks for the comment. The proposed hidden graph features are learned hierarchical representations that jointly capture both structural and attribute information in a unified embedding space. By computing kernel-based similarities between input graphs and a set of trainable anchor graphs across multiple hierarchical levels, HSP-GKN encodes more task-relevant graph semantics in a soft and adaptive manner. Specifically, for each input graph, a similarity vector is constructed by measuring the similarity between its hierarchical features $\phi_l(G)$ and the learned features of the anchors at each level $l$. Each element of this vector reflects the similarity at a specific level of abstraction. The resulting vector serves as the final graph-level representation for downstream tasks, enabling the model to explicitly preserve and utilize structural and attribute information from multiple levels of hierarchy.
>
> **Question 3:** **Comparison with SP kernel** Thanks for the comment. SP kernel compares each pair of shortest paths of the same and different lengths between two graphs. Consider two shortest paths $\pi_a$ and $\pi_b$ with lengths $l$ and $l+1$ respectively. Their overlapping nodes share identical labels, the endpoint label of $\pi_a$ differ from those of $\pi_b$. In this case, the SP kernel produces counterintuitively low similarity scores between the paths. Therefore, although the shortest-path kernel allows for comparisons between paths of different lengths, such comparisons often fail to accurately reflect the true similarity between the paths in most cases. Furthermore, by restricting comparisons to paths of equal length, we enable the construction of explicit feature mappings, which significantly reduces the overall time complexity. This also lays the foundation for designing an end-to-end trainable framework.
>
> **Weaknesses 1&5** Thanks for your comment. Existing GKNN methods typically rely on predefined graph kernels and perform downstream optimization based on the resulting similarity measures. However, these approaches generally do not modify the original kernel computation, and their scalability is inherently limited by the chosen kernel. In this paper, we propose a computationally efficient explicit feature mapping that integrates both structural and semantic information, enabling graphs to be mapped into a vector space suitable for neural network processing. We will give more discussion about the differences between the proposed approach and existing GKNNs and kernel methods in the revision.
>
> **Weaknesses 2** Thanks for your comment. Indeed, using a linear kernel makes HSP less flexible. However, restricting the path kernel to a linear kernel brings additional benefits. On the one hand, it significantly reduces the computation time of the kernel. On the other hand, it allows for the construction of explicit feature mapping, making it easier to integrate downstream tasks into an end-to-end framework. Furthermore, with explicit features, an MLP can be used to capture nonlinear relationships. We will discuss this further in the revision.
>
> **Weaknesses 3** **Datasets details** Thanks for your comment. Following your suggestions, we will add more detailed description of the datasets in the final paper.
>
> [1] Agent-based graph neural networks. ICLR, 2023.

---

> ### Comment · Reviewer_y3Cv · 2025-08-03
>
> I appreciate the authors' thoughtful rebuttal. Their responses have addressed my concerns. Accordingly, I recommend accepting the paper.

---

> > ### Author Response · Authors · 2025-08-04
> >
> > Dear Reviewer y3Cv,
> >
> > We sincerely thank you for the detailed and valuable comments on our paper.
> >
> > Best regards,
> >
> > Authors of Paper 8752

---

### Note · Authors · 2025-08-16

Dear Area Chair and Reviewers,

We express our sincere appreciation for the insightful remarks made by the reviewers that enhance the quality of our manuscript. We would like to highlight the key improvements that were made based on the recommendations of the reviewers.

**Technical Contributions:** We clarify the innovation of HSP-GKN for Reviewer **wTHL** compared to prior works such as DHAKR and GKNNs. Unlike count-based or fixed-kernel approaches, our method introduces an explicit hierarchical feature mapping combined with learnable hidden graph features, enabling task-adaptive similarity-based representations in a continuous embedding space. This design ensures improved expressiveness and scalability while retaining both structural and attribute semantics.

**Experimental Validation:** We extend experiments to large-scale benchmarks (OGBG-molhiv, REDDIT-M-12K) as mentioned by Reviewers **y3Cv** and **wTHL**, where HSP-GKN consistently outperforms existing kernels and GNNs while remaining computationally efficient. Additional comparisons with state-of-the-art kernels request by Reviewer **QL9x** further validate the strong performance of our framework. Detailed dataset descriptions and corrections to typos and figures will also be incorporated in the revision for better clarity and reproducibility.

**Future Extensions:** We carefully consider the suggestions of Reviewer **wTHL** on extending HSP-GKN to graphs with edge attributes. While most benchmark datasets used do not contain edge features, we have discussed how to incorporate edge attributes to better support the domains, such as molecular property prediction.

The positive evaluations from Reviewers **y3Cv**, **QL9x**, **3EPr**, and the constructive comments from Reviewer **wTHL** affirm the technical quality and significance of our contribution. We believe that the reviewers have further understood and recognized our paper.

Best regards,

Authors of Paper with ID 8752

---

### Decision · Program_Chairs · 2025-09-17

**Decision:**

Accept (spotlight)

**Comment:**

The paper introduces a novel Graph Kernel, the Hierarchical Shortest-Path Graph Kernel, designed to be used as a convolutional layer within a graph neural network in an end-to-end optimization framework. Building on prior work on kernelized GNNs, the main contribution lies in a computationally efficient shortest-path kernel that allows for practical scalability. The experimental evaluation demonstrates strong performance and runtime improvements across a range of benchmarks.

One reviewer raised concerns about the contribution being incremental and about the lack of edge attribute support, which could limit applicability in certain domains. However, the authors pointed out that this limitation is common to many kernel-based methods, which still offer an alternative and interesting perspective on graph convolutions. Furthermore, the introduction of a differentiable and efficient kernel, combined with favorable empirical results, represents a meaningful advance in this line of research. Most reviewers acknowledged the novelty and technical soundness of the approach and considered it a valuable contribution.